# Design Optimization and Carbon Footprint Analysis of an Electrodeionization System with Flexible Load Regulation

Yuan Yuan [1,†], Fengting Qian [1,†], Jiaqi Lu [1,*], Dungang Gu [1], Yuhang Lou [1], Na Xue [1], Guanghui Li [1,*], Wenjie Liao [2] and Nan Zhang [3]

1 Innovation Centre for Environment and Resources, School of Chemistry and Chemical Engineering, Shanghai University of Engineering Science, No.333 Longteng Road, Songjiang District, Shanghai 201620, China
2 Institute of New Energy and Low-Carbon Technology, Sichuan University, Chuanda Road, Shuangliu County, Chengdu 610207, China
3 Centre for Process Integration, Department of Chemical Engineering and Analytical Science, The University of Manchester, Manchester M13 9PL, UK
* Correspondence: wilsherelu@foxmail.com (J.L.); ghli@sues.edu.cn (G.L.); Tel.: +86-21-67795965 (J.L. & G.L.)
† These authors contributed equally to this work.

**Abstract:** Thermal power plants will function as a flexible load regulation in a low-carbon grid, which requires operation adaption for the whole system. Energy transition in the electricity sector is the core to realizing carbon neutrality. The power grid will be gradually dominated by renewable energy, such as wind power and photovoltaic solar power. However, renewable energy has problems such as insufficient power supply and output fluctuation; thermal power will be required to regulate the peak load flexibly to meet demand. Therefore, the supply of boiler make-up water prepared by electrodeionization (EDI) in thermal power plants should also be flexibly changed. This study focused on the ultrapure water preparation system by EDI with variable flow rates. For an EDI system with a maximum ultrapure water capacity of 20 m$^3$/h, the power consumption, annual cost, and carbon footprint of different designs are compared. The operation parameters were optimized based on the optimal cost design when the temporal demand of boiler make-up water is reduced to 75%, 50%, and 25%, respectively, considering thermal power as peak load regulation technology. The results showed that the optimized system could significantly reduce power consumption and carbon footprint by up to 30.21% and 30.30%, respectively. The proposed strategy is expected to be widely applied for design and operation optimization when considering the low-carbon but unstable energy system dominated by renewable energy. The carbon footprint could be a feasible optimization object to balance the greenhouse gas emissions from the module manufacturing and operation consumption.

**Keywords:** ultrapure water; peak load regulation; electrodeionization model; optimization; life cycle assessment

## 1. Introduction

To mitigate ever-increasing global climate change, all countries signed a historic United Nations climate agreement in Paris to jointly tackle greenhouse gas (GHG) emissions [1–3]. China also pledged to reach its peak of carbon dioxide emissions by 2030 and achieve carbon neutrality by 2060 [4]. According to the International Energy Agency report, GHG emissions of the energy and electricity sectors had reached their highest value, and the growth between 2019 and 2021 came almost entirely from China [5]. Hence, the decarbonization of the energy and electricity sectors is the priority for achieving carbon neutrality, especially in China. Currently, the energy structure in China is still dominated by fossil energy, particularly coal. Energy transition is the inevitable pathway to reduce the gross amount and intensity of GHG emissions, and fossil energy should be replaced by renewable energy to reduce the overall GHG emissions of society. In the future, non-fossil energy, especially wind and photovoltaic solar power, will become the majority. These primary energy

sources will mainly be converted into electric energy for utilization [6]. In the global energy-structure transition process, due to the increase in power load and the sudden drop in renewable energy output [7], some countries have experienced short-term power supply shortages, resulting in power rationing or unplanned power outages. Therefore, the issue of power supply stability under a low-carbon grid should be emphasized. One of the countermeasures is to increase the flexibility of the electricity system, such as regulating the peak load by thermal power to increase the capacity of the system to absorb wind power and photovoltaic power. The production and consumption of electric energy should be balanced in real time. When a large amount of unstable renewable energy power dominates a grid with limited storage capacity, thermal power plants should function as a load regulation to cut the demand peak. When the output of renewable power is high, the thermal power will run at a low load, and vice versa. Therefore, thermal power plants should flexibly operate at different loads (flexible load regulation).

Most of the current thermal power generation in China is based on coal combustion. The chemical energy in the fuel is converted into thermal energy and absorbed by the water in the boiler. The liquid-state water is evaporated into superheated steam with sufficient quantity and certain qualities (steam temperature and pressure) for power generation by the steam turbine [8]. The water should be supplemented by the boiler (so-called boiler make-up water) due to the loss due to discharge, evaporation, leakage, or extraction. The regulated conductivity, pH, and ion concentration of boiler make-up water are close to those in the standard of ultrapure water (UPW). Otherwise, the related pipelines and equipment undertake a risk of damage. The preparation of UPW by integrated membrane technology (IMT) has gradually replaced the traditional ion exchange technology (IX) [9] since IMT technology facilitates efficient ion removal without chemicals for resin regeneration and ensures the quality and continuous preparation of UPW. Electrodeionization (EDI) technology is a polishing process in IMT for achieving the desired water quality.

Compared with using multiple modules with low flow rates (small modules) to achieve the same effect, modules with high flow rates (large modules) can save the raw materials such as module shells and electrodes. Therefore, considering the cost, industrial UPW preparation by EDI mainly uses large modules. However, under the background of the energy transition, the proportion of thermal power generation in power consumption will decrease. Meanwhile, the boiler make-up water required for thermal power generation will also be flexibly reduced given thermal power as a peak load regulation. Therefore, using small modules facilitates the EDI system to change the volume of boiler make-up water produced. In any case, it is still necessary to establish a real-time scheduling model for the flexible operation.

At present, there is much research related to the process optimization of water treatment membrane technologies [10–14]. However, most of the aforementioned research is based on experimental approaches without mathematical modeling. This paper applied the mathematical process model for optimizing an EDI system. Although the modeling of the EDI process has been extensively studied [15–19], most of these studies were established to explain the mechanism of the EDI process without optimization and the dissociation of water at the membrane interfaces was absent. Herein, the operation and design (number of dilute chambers, width of dilute chambers, membrane area, etc.) of the EDI process were simultaneously optimized with a mechanism-based model including water dissociation.

Meanwhile, GHG emissions also need to be evaluated for an industrial process to support the systematic revolution toward carbon neutrality. Life cycle assessment (LCA) is a widespread, effective, and systematic analysis method to quantify the resource consumption and environmental impact throughout the life cycle of products, processes, or activities [20–23]. LCA can be used to quantify the carbon footprint (i.e., life-cycle GHG emissions) under the conditions of the current industry and identify the main source of environmental impacts (hotspots) for further improvement. In addition, the emission reduction potential of carbon-neutral technologies can be revealed, making it possible to put forward the optimal design [24–27]. At present, there are some LCA studies on water

treatment processes such as multi-stage flash [28], low-temperature multiple-effect distillation [28], nanofiltration [29], and reverse osmosis [30,31]. However, none of the current studies have considered a flexible operation with an unstable energy system dominated by renewable energy.

The purpose of this study is to suggest the low-carbon design and operation of the EDI system, considering the flexible load regulation of thermal plants. This study applied a two-dimensional finite element model of the UPW preparation process by EDI established in our previous work [32]. First, EDI technology and the EDI system are introduced in Sections 2.1 and 2.2, and the modeling method of EDI systems and modules are described in detail in Sections 2.3 and 2.4. A sensitivity analysis is carried out to verify the model in Section 3.1. Then, the design and operation of the EDI system are optimized under different circumstances in Sections 3.2 and 3.3, with a carbon footprint analysis in Section 3.4. Finally, the refined suggestions are put forward in Section 3.5. The results of this study could guide the optimal design and operation of the EDI system considering flexible load regulation, and the derived carbon footprint analysis could also provide quantitative evidence for the low-carbon design of EDI modules. The strategy proposed in this study could be applied to the design and operation optimization of industrial processes integrated with unstable renewable energy.

## 2. Case Study and Methods

### 2.1. Principle of UPW Preparation by EDI

EDI is an electrically driven desalination technology. The EDI module is composed of positive and negative electrodes and alternately arranged dilute chambers and concentrated chambers, formed by the alternately arranged ion exchange membranes. The dilute chambers are filled with ion exchange resins, as shown in the Supporting Information (SI), Figure S1. The anionic and cationic ions in the influent exchange with the ion exchange resin in the dilute chamber, then migrate to the electrodes through the ion exchange resin with the direct current (DC). The anions migrate to the anode, the cations to the cathode, and finally enter the adjacent concentrated chambers through the ion exchange membranes on both sides of the dilute chamber [33]. At the same time, under the action of a DC electric field, the water molecules in the dilute chamber disassociate into $H^+$ and $OH^-$. Without additional acid and alkali, the ion exchange resin can be regenerated by the $H^+$ and $OH^-$ generated by the disassociation of the water molecules [34]. Therefore, EDI is generally considered a green and environmentally friendly technology.

### 2.2. EDI System

A typical EDI system for UPW preparation contains multiple modules, and the total industrial demand for UPW usually determines the number of modules required.

For a brief description, the module with a flow rate of 4 $m^3/h$ can be set as the benchmark. As shown in Figure 1, each module's water flow is parallel, and the current between each module is in a series. For the EDI system composed of modules with a flow rate of 2 $m^3/h$, as shown in Figure 1a, two times the number of modules is necessary compared to 4-$m^3/h$ modules, as shown in Figure 1b. The water flow direction of 2-$m^3/h$ EDI modules in each pair is parallel, and the current is in a series. The current among different pairs is parallel. The total current equals the sum of the partial currents that pass through each parallel connection.

Disregarding peak load regulation, it is assumed that all the power required is provided by the thermal power generation. The demand for boiler make-up water in the thermal power plant is 20 $m^3/h$. For the EDI system, three design schemes for the 20-$m^3/h$ EDI system are carried out, respectively: (1) the system is only composed of large modules (3.4–4.5 $m^3/h$); (2) the system is only composed of small modules (1.5–2.5 $m^3/h$); and (3) the EDI system is freely composed of large and small modules.

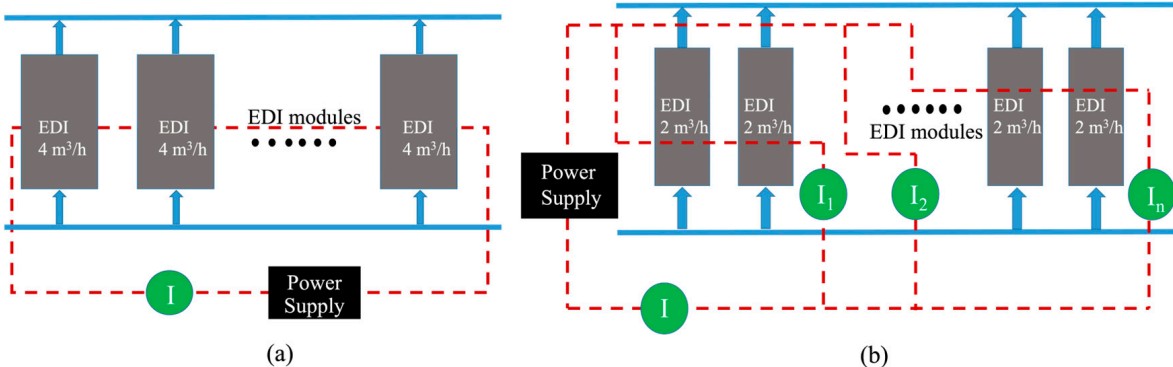

**Figure 1.** Schematic of the EDI system: (**a**) composed of large flow modules; (**b**) composed of small flow modules. Blue lines indicate the water flow direction; red dotted lines indicate the current direction.

### 2.3. Modeling of EDI Modules

The schematic diagram of an EDI module is shown in Figure 2.

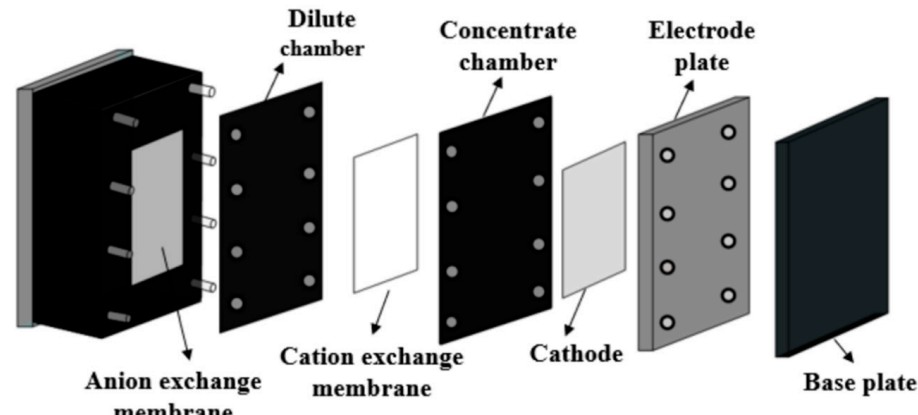

**Figure 2.** Schematic diagram of the internal structure of an EDI module.

It is necessary to determine the optimal design of the EDI system with the lowest energy consumption and cost to meet the default demand of boiler make-up water. In addition, to realize the flexible UPW preparation under different demands, the operation optimization of each EDI module is also essential to minimize energy consumption. Consequently, both the design and operation of an EDI system should be optimized by a non-linear mathematical programming algorithm to minimize the operation energy consumption and economic cost. In this study, a E-Cell MK-2 EDI module on the market was selected for the modeling and optimization. The relevant parameters of this module are shown in Table 1.

**Table 1.** Operation and design parameters of the E-Cell MK-2 module.

| $A$ (cm²) | $L$ (cm) | $d$ (cm) | $W$ (cm) | $F$ (m³/h) | Number of Dilute/Concentrate Chambers | $q_i$, $q_j$ (mmol/mL) | $K_i$, $K_j$ (-) |
|---|---|---|---|---|---|---|---|
| 507 | 39 | 13 | 0.8 | 3.5 | 36/37 | 2 | 1.55 |

In our previous study, based on a dilute chamber in the EDI module, a process model of UPW preparation by EDI was established in GAMS (the General Algebraic Modeling System) with a two-dimensional finite element method. The detailed mechanism of the

process model is included in the Supporting Information, Section S3, and more modeling details can be found in our previous article [32].

### 2.4. Construction of the EDI System Optimization Model

In addition to the operation variables such as flow rate ($F$) and current ($I$) in the model of the EDI system, there are also some design variables for the modules: the ion exchange membrane area ($A$), membrane length ($L$), and width ($d$), dilute chamber width ($W$), and the number of diluter chambers ($N_d$), and concentrated chambers ($N_c$). The upper and lower variable limits are about $\pm 10\%$ to achieve optimization within a reasonable range of the model design. There are two aspects of the optimization objectives to meet the standard quality of produced UPW: (1) the simultaneous design optimization of an operation under a given maximum UPW production flow, fulfilling the primary objective of minimizing the total costs of operation and design (shown in Equation (3)); and (2) operation optimization under flexible scheduling, fulfilling the objective of minimizing power consumption during operation (shown in Equation (2)).

Objective function (1):

$$\text{Minimize } I^2 \times R_{\text{module}} \times N_{\text{module}} \tag{1}$$

Operating costs caused by power consumption:

$$I^2 \times R_{\text{module}} \times N_{\text{module}} \times \text{Cost}_E \times T_H \tag{2}$$

The design cost includes the following four aspects:

(1)  Ion exchange membrane: $(N_d + N_c + 1) \times A \times N_{\text{module}} \times \text{Cost}_M$
(2)  Ion exchange resin: $N_d \times A \times W \times N_{\text{module}} \times \text{Cost}_R$
(3)  Enclosed shell: $2 \times (A + W \times L + W \times d) \times N_{\text{module}} \times \text{Cost}_{\text{shell}}$
(4)  Electrodes: $2 \times A \times N_{\text{module}} \times \text{Cost}_{\text{ele}}$

Objective function (2):
Minimization of annual cost:

$$
\begin{aligned}
&= \frac{h(h+1)^g}{(h+1)^{g-1}} \times N_{\text{module}} \times [(N_d + N_c + 1) \times A \times \text{Cost}_M \\
&+ N_d \times A \times W \times \text{Cost}_R + 2 \times \text{Cost}_{\text{shell}} \times (A + W \times L + W \times d) \\
&+ 2 \times A \times N_{\text{module}} \times \text{Cost}_{\text{ele}}] + I^2 \times R_{\text{module}} \times N_{\text{module}} \times \text{Cost}_E \times T_H
\end{aligned}
\tag{3}
$$

Restricted condition:

$$C_{\text{out}}(i) \ll C^2_{\text{out}} \tag{4}$$

$$F_{\text{lo}} \leq F \leq F_{\text{up}}; I_{\text{lo}} \leq I \leq I_{\text{up}} \tag{5}$$

$$A_{\text{lo}} \leq A \leq A_{\text{up}}; W_{\text{lo}} \leq W \leq W_{\text{up}} \tag{6}$$

### 2.5. Life Cycle Assessment of Optimization of the EDI System

LCA was also applied to quantify the reduction effect of the carbon footprint for the optimized EDI system. In this study, based on ISO 14040 series guidelines [35], the GHG emissions associated with the manufacturing, operation, and waste disposal of an EDI module, as shown in Figure 3, were researched. To evaluate such cradle-to-grave impacts, the functional unit is defined as the production of 1 metric ton (MT) of UPW by EDI.

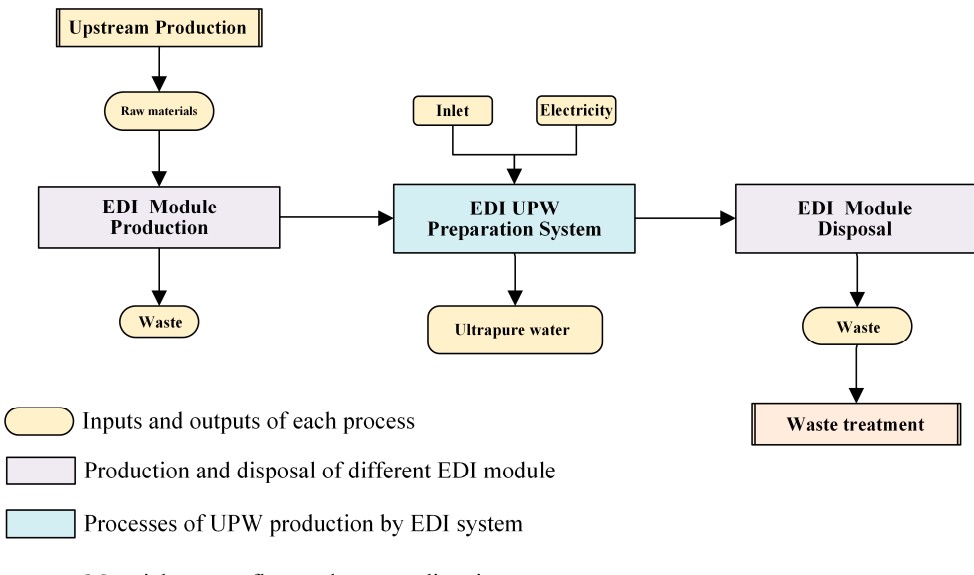

**Figure 3.** System boundary for LCA of EDI system.

The lifespan of EDI modules is assumed to be 5 years, and the size and specification of the EDI modules were used to obtain the manufacturing and waste management inventory data, as well as the operating power consumption based on the optimization of operation parameters. The carbon footprint of the optimized EDI system was quantified according to the global warming potential (GWP) values of a variety of GHGs reported by the Intergovernmental Panel on Climate Change (IPCC) in 2013. The IPCC's 100-year GWP method normalizes the various GHGs into the weight of carbon dioxide based on the respective absorption capacity of infrared radiation within a 100-year time frame, which is commonly used to measure impacts on climate change [36]. The detailed inventory data and calculation process are included in the Supporting Information, Sections S5–S11.

## 3. Results and Discussion

In this chapter, a sensitivity analysis on the operation and design variables of the EDI system with different module combinations was conducted to understand and validate the established model. Then, the design of the 20-m$^3$/h EDI system was optimized to obtain the lowest economic cost design. Fixing this optimal design, the operation of the EDI system was optimized to make the EDI system be able to flexibly schedule under changes of boiler make-up water demand, and LCA was carried out to analyze the carbon footprint of EDI systems. Finally, the results were discussed, and suggestions were put forward.

### 3.1. Sensitivity Analysis

To validate the effectiveness of the established model for the EDI process, a sensitivity analysis was conducted for the 20 m$^3$/h EDI UPW preparation system composed of 5 large modules. When the module number is fixed, a change of optimized results due to the change of UPW demand is studied. The operation and design parameters at the lowest annual cost can be calculated through the optimization model with the variation of the total flow rates. The sensitivity analysis of optimization results under different boiler make-up water demands is shown in Figure 4 and Supporting Information, Table S3.

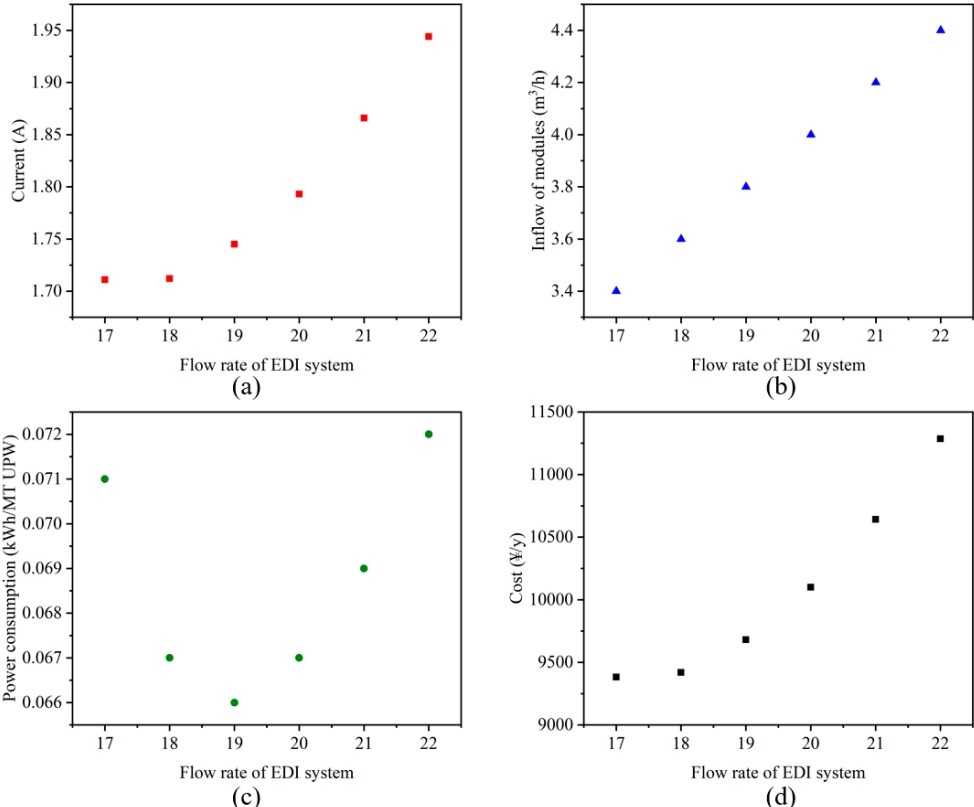

**Figure 4.** Sensitivity analysis of UPW demand: (**a**) current with different system flow rates; (**b**) inflow of modules of different systems with different system flow rates; (**c**) power consumption with different system flow rates; (**d**) cost with different system flow rates.

By comparing the results in Figure 4a,b, it can be seen that when the flow rate of the EDI system increases, the water treatment capacity of a single module increases, causing the rise of the operating current, ultimately increasing annual power consumption and cost as the EDI system flow increases. Figure 4c shows that with the increase of EDI system flow, the power consumption of MT UPW produced presents a reverse parabola trend that decreases first and then increases. The increase of flow rate is a key factor in reducing the dissociation of water in the EDI process [37]. At a low ion concentration and low flow rate, there are insufficient ions at the interface of the ion exchange membrane, and the dissociation of water in the EDI process is strengthened [38], resulting in reduced current efficiency. Therefore, when the flow rate is less than 19 $m^3$/h, energy consumption is relatively high. Compared with the experimental data in the literature [15], the variation trend of variables in the sensitivity analysis conforms to the experimental data; thus, the model is robust.

When the UPW demand is less than 19 $m^3$/h, the current of modules increases slightly with the increase of UPW demand. This is because the flow rate varies in the range of 3.4–4.5 $m^3$/h, and the cost of the operating power consumption is much higher than the module cost. The operating current is low when the flow rate is low, while the contribution of the power consumption cost to the total cost declines. Therefore, to minimize the total cost, the reduction of the operating current is the first priority.

The sensitivity analysis of the membrane area was also conducted and is included in the Supporting Information, Section S2. The results show that for the same boiler make-up water demand, an increase of the membrane area can reduce the operating current, thereby reducing both the power consumption per MT of UPW produced and annual cost (Table S4). Therefore, the upper limit of the membrane area is defined in this study. The membrane area of the EDI model in this study is 600 $cm^2$ and the width of the dilute chamber is 0.6 cm.

### 3.2. Design Optimization of 20-m³/h UPW Preparation System by EDI

As mentioned in Section 2.2, a 20-m³/h EDI system can be composed of large modules, small modules, or a combination of large and small modules. Table 2 shows that in the three designs, the power consumption per MT of UPW produced is similar. Because the cost mainly comes from the operating power consumption, the most energy-efficient design and operation were emphasized.

**Table 2.** Design optimization of 20 m³/h EDI system with different module specifications.

| UPW Demand (m³/h) | 20 | | | |
|---|---|---|---|---|
| System | Inflexible System | Small System | Flexible System | |
| Module number | 5 large | 8 small | 4 large | 1 small |
| Inflow (m³/h) | 4 | 2.5 | 4.2 | 2.4 |
| $I$ (A) | 1.79 | 1.791 | 1.786 | 1.787 |
| $A$ (cm²) | 600 | 600 | 600 | 600 |
| $d$ (cm) | 0.6 | 0.6 | 0.6 | 0.6 |
| $N_d$ | 38 | 24 | 42 | 24 |
| Power consumption (kWh/MT UPW) | 0.067 | 0.067 | 0.067 | |
| Annual cost (¥/y) | 10,200.2 | 10,457.7 | 10,101.1 | |

When an EDI system is composed of small modules, the number of small modules can range from 8–13. When there are 8 small modules, the cost is the lowest, because when only small modules are used, the number of EDI modules required for the system increases, resulting in a higher investment cost. It can also be seen that when an EDI system is composed of small modules, the operating current of a single module is the largest among the three designs.

When large modules are combined with small modules, the inflow of large modules increases, and the number of dilute chambers of large modules must be increased to meet production quality, meaning a higher investment cost. However, when large modules are combined with small modules, the operating current can be reduced, meaning a lower operation cost.

The results show that the EDI system composed of large and small modules has the most advantage in terms of cost. This is because, compared with the EDI system composed of large modules, the use of small modules reduces the consumption of raw materials which reduces the investment cost. However, the combination system operates at the maximum flow rate within the modules' flow range, which may lead to a shortened lifespan of the modules; this was not considered in this study.

### 3.3. Energy-Saving Effect of Operation Optimization under Flexible Peak Load Regulation

Considering an EDI system in a thermal power plant with a flexible peak load regulation, the demand for boiler make-up water will face all possible circumstances. The EDI system design with the minimal annual cost is composed of 4 large modules and 1 small module (flexible system), as shown in Section 3.2. Based on this design, the operating parameters were optimized to flexibly adapt to a variable UPW demand under peak load regulation. The demand is reduced to 75% (15 m³/h), 50% (10 m³/h), and 25% (5 m³/h) for quantifying the optimal operating variables in the EDI system with the lowest energy consumption. Meanwhile, as a contrast, the EDI system only composed of large modules was also analyzed.

The energy-saving effect of operation optimization under flexible peak load regulation is shown in Figure 5a. When the UPW demand is 15 m³/h, in the inflexible system the operating current of a single module is 1.73 A for the EDI system with 5 large modules, and in the flexible system the operating current of the large modules and small module is

1.75 A and 1.74 A, respectively. Although the operating current in the inflexible system is relatively low, the modules operate at a relatively low flow rate, resulting in a low current efficiency. At this time, the power consumption per MT of UPW produced by the two systems is very close.

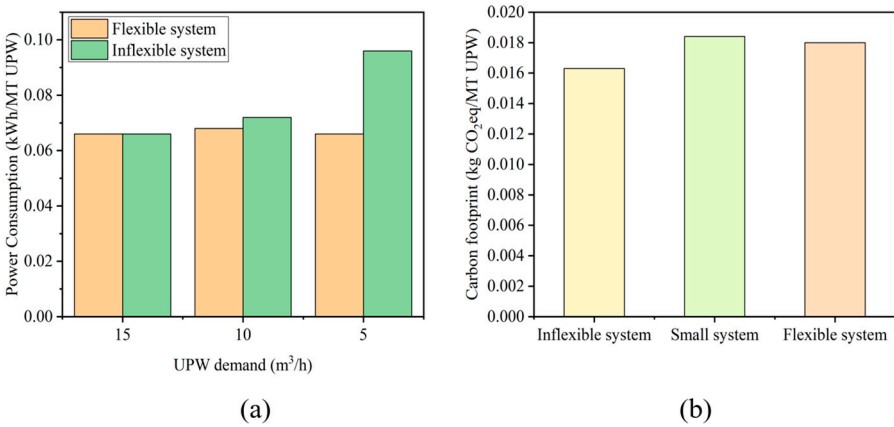

(a)        (b)

**Figure 5.** The comparison of flexible and inflexible systems: (**a**) power consumption under different UPW demands; (**b**) carbon footprints of different systems with a 20 m$^3$/h UPW demand.

When the UPW demand drops to 10 m$^3$/h, the operating current of a single module in both systems is 1.71 A. However, the power consumption per MT of UPW produced by the inflexible system is higher than that of the flexible system. Because three large modules are required to meet the 10 m$^3$/h demand for UPW, the minimum flow rate is 3.75 m$^3$/h. The produced UPW is greater than the demand, resulting in a waste of produced UPW. Meanwhile, the low current efficiency caused by the low flow rate increases the power consumption.

When the UPW demand is 5 m$^3$/h, the operating current of a single module is 1.71 A in the inflexible system, and the operating current of a single large and small module is 1.75 A and 1.74 A in the flexible system, respectively. This is similar in the case of 10 m$^3$/h, where the power consumption of the flexible system is much lower than that of the inflexible system for the same reason.

Based on the findings in this section, it can be concluded that the flexible EDI system can avoid unnecessary UPW production waste as the UPW demand changes, especially for low flow rates which the large modules cannot exactly reach. In addition, when the modules operate at a higher flow, the current efficiency is higher so that more energy can be saved.

### 3.4. Comparative Analysis of Carbon Footprint for Different Designs

This study quantified the carbon footprint of EDI systems with different designs according to the corresponding inventory data, background data sources, and calculation method included in the Supporting Information, Sections S5–S11. The carbon footprints of the different designs of the EDI systems are shown in Figure 5b. When the UPW demand is 20 m$^3$/h, the flexible system has a higher carbon footprint compared with the inflexible system. It can be seen from Section 3.2 that for the 20-m$^3$/h EDI system, the power consumption per MT of UPW produced by the three designs is the same. Thus, the carbon footprint caused by operating power consumption is close. Meanwhile, large modules in the flexible system have a greater number of dilute chambers, resulting in more raw material consumption during module production and more waste that needs to be treated.

It is difficult to know the life cycle operation time of the EDI system for different flow rates under flexible peak load regulation. Thus, the life cycle carbon footprint cannot currently be evaluated. Instead, according to the calculated operating power consumption in Section 3.3, the carbon footprint during operation can be quantified to estimate the GHG reduction potential owing to the flexible EDI system. Based on the present annual

power generation by power plants in China, the flexible EDI system could annually reduce 127,000 MT of GHG emissions compared with the inflexible system when the thermal power load is reduced to 25%.

For operation at constant flow rate, the carbon footprint of the flexible system is 9.56% higher than that of the inflexible system. However, considering flexible scheduling operation, the GHG emission reduction brought about by reduced operating power consumption has great potential to cover the additional emissions from the production of the modules. Therefore, the transition to a flexible-scheduling EDI system is necessary.

### 3.5. Further Discussion and Suggestions

Due to the lack of relevant data, this study did not calibrate the model parameters for large and small modules. Further research could be conducted to distinguish the model with different module specifications. The model of the EDI process in this study only considers the dilute chamber without the diffusion phenomenon. Further studies of the model could include the concentrate chamber and diffusion, and locate the model closer to its actual industrial operation. At the same time, this paper lacks the carbon footprint calculation method under flexible scheduling, which requires the total UPW yield according to the proportion of different UPW production flow rates. Although there are limitations, the results of this study still have reference value for the membrane treatment process. There have been some studies on the membrane water treatment process and renewable energy system [39–41], but few studies have mentioned the influence of the instability of the renewable energy system on the membrane water treatment process and optimized the membrane water treatment process on this basis. This study focused on the impact of the instability of the renewable energy system on the process of boiler make-up water preparation by EDI, and obtains some valuable conclusions.

For different boiler make-up water demands, large modules are crucial to reduce the cost in EDI systems. The results in Section 3.1 show that in the design of large modules, the increase of membrane area will increase the cost of the device and can significantly decrease the operating power consumption. Thus, the reduced cost brought by the energy-saving is more significant than the additional cost of the increased consumption of raw materials. Ultimately, the total annual cost is more competitive.

However, when thermal power functions as the peak load regulation, small flow modules in the flexible system can enable the EDI system to extensively adapt to variable UPW demand and avoid excess production. Moreover, the smaller the demand for UPW is, the more noticeable the energy-saving is. Therefore, to reduce the power consumption and carbon footprint of EDI operation, a combination of small and large modules in the EDI system is necessary when the system needs to be flexibly scheduled.

### 4. Conclusions

Based on the optimization and LCA results, the necessity of promoting the flexibility of the boiler make-up water supply was demonstrated under the carbon-neutral revolution of thermal power plants. For the design of an EDI system operating at a fixed flow rate, the most economic design is the combination of large and small modules. Still, additional GHG emissions will occur because of the greater amount of materials required for the manufacturing of the module. When thermal power plants need to work for peak load regulation, the boiler make-up water demand should be flexibly changed. In this situation, the combination of small and large modules in the EDI system can reduce both the cost and the operating power consumption, and the corresponding GHG emissions from power consumption will also decrease.

When the flexible EDI system is applied, the operating power consumption is reduced by up to 30.21% compared to the regular design, and the flexible system may annually reduce 127,000 MT of GHG emissions in China. Therefore, the combination of small and large modules is critical and essential for EDI operation with flexible scheduling. For a common membrane water treatment process, firstly, the water product flow rate can

be scheduled according to the real-time output of renewable energy. In consequence, the peak of renewable energy output can be flexibly consumed by the membrane water treatment process. Then, when electricity becomes low-carbon, GHG emissions from process electricity consumption will be greatly reduced. At that time, it will be necessary to consider low-carbon membrane production and waste treatment. This study could provide a valid tool for optimizing the EDI system in power plants with boiler make-up water supplies, and the proposed strategy could be extended for other water treatments or industrial processes to obtain the optimal design and operation when the instability of renewable energy needs to be considered. The model parameters should be further adjusted based on the different specifications of EDI modules; meanwhile, the concentrate chamber and ion diffusion could be included in the model. Moreover, future research needs to quantify the running time of the flexible-scheduling EDI system under different UPW demands for the carbon footprint calculation.

**Supplementary Materials:** The following supporting information can be downloaded at: https://www.mdpi.com/article/10.3390/su142315957/s1 [32,42,43].

**Author Contributions:** Y.Y.: Data curation, Writing—Original draft preparation. F.Q.: Investigation, Software, Writing—Original draft preparation. J.L.: Methodology, Software, Writing—Review & Editing. D.G.: Validation. Y.L.: Software. N.X.: Data Curation. G.L.: Project administration, Supervision. W.L.: Writing—Review & Editing. N.Z.: Conceptualization, Supervision. All authors have read and agreed to the published version of the manuscript.

**Funding:** This work was supported by Capacity Building Project of Some Local Colleges and Universities in Shanghai (No. 21010501400).

**Conflicts of Interest:** The authors declare no conflict of interest.

## Abbreviations

| | |
|---|---|
| $A$ | Membrane area ($cm^2$) |
| $C$ | The concentration of the liquid phase ($Equiv/cm^3$) |
| $Cost_E$ | Electricity fee (¥/kWh) |
| $Cost_M$ | Price of ion exchange membrane (¥/$cm^2$) |
| $Cost_R$ | Price of ion exchange resin (¥/L) |
| $Cost_{shell}$ | Price of ion shell (¥/$cm^2$) |
| d | Width of the membrane (cm) |
| $F$ | The flow rate of dilute chamber ($cm^3/s$) |
| $F_c$ | The flow rate of each cell ($cm^3/s$) |
| $g$ | The lifespan of EDI (year) |
| $h$ | Interest rate |
| $i$ | Cation |
| $I$ | Current (A) |
| $I_{line}$ | Row current (A) |
| $j$ | Anion |
| $L$ | Membrane length (water flow direction) (cm) |
| lo | Variable lower limitation |
| $m$ | Number of columns |
| $N_d$ | Number of dilute chambers |
| $N_c$ | Number of concentrate chambers |
| $n$ | Number of rows |
| $PC$ | Energy consumption per ton of water treated ($Wh/m^3$) |
| $R$ | Membrane stack resistance ($\Omega$) |
| $T_H$ | Annual working time (h) |
| up | Variable upper limitation |
| $w$ | Width of the dilute chamber (current direction) (m) |

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
