# Peer review of "Design Optimization and Carbon Footprint Analysis of an Electrodeionization System with Flexible Load Regulation"

_sustainability, doi:10.3390/su142315957_

Round 1
Reviewer 1 Report
The article entitled “Design optimization and carbon footprint analysis of an electrodeionization system with flexible load regulation” is written very and according to the journal's scope. I want to recommend changes before the final acceptance of the article.
Major revision:
1. Please start the abstract with the main objectives of the study.
2. I recommend you to add more relevant literature [1,2] with the first sentence of the introduction as “To mitigate the ever-increasing global climate change, all countries signed a land-36 mark United Nations climate agreement in Paris to jointly tackle greenhouse gas (GHG) 37 emissions [1,2]”
[1] The Paris Climate Agreement and China's Role in Global Climate Governance
[2] Extreme weather events risk to crop-production and the adaptation of innovative management strategies to mitigate the risk. Technovation. Volume 117, 102255
3. What are the flexible load regulations? Please clearly define or explain in the second paragraph of the introduction.
4. Please write main research questions and contributions of the study at the end of the introduction.
5. It would be easier for the readers to follow the paper if you could add the article's structure at the end of the introduction. Moreover, please explain the section work.
6. In the whole section on results and discussion, I highly recommend justifying each finding with previous studies and adding updated literature while discussing or comparing the findings.
7. Please add the main limitations and recommendations for future studies at the end of the conclusion.
Reviewer 2 Report
Dear Authors,
The topic of this article is very important from the scientific and practical points of views and are related to design optimization and carbon footprint analysis of an electro-deionization system.
There are some comments for the improvements of this article:
1. This statement from Abstract is very general “The proposed strategy is expected to be widely applied for design and operation optimization...“, and Authors can adjust some details according to different situations with membranes treatment plants.
2. When reading Introduction part, there are statements on main novelty of this research missing by the end of different methods.
3. All methods were precisely written and discussed with all related details and estimation formulas.
4. Some Figures in Results and Discussions are not informative, e.g. Figure 4 and Figure 6. It is better, if Authors will combine all related results in one comprehensive Figure.
5. Conclusions are not fulfilled with recommendations to membranes water treatment plants operators, how we can improve it?
6. References can be added with more global research on membranes techniques, (e.g. "Fluoride Removal from Groundwater by Technological Process Optimization") and/ or similar research.
Sincerely, Reviewer.
Round 2
Reviewer 1 Report
The athors have addressed all previous comments, and now it is acceptable for publication in its present form.